# Reconceptualizing the Study of Christian Universities in the Republican Era in Today's China

Peter Tze Ming Ng

Chinese University of Hong Kong, Shatin, Hong Kong SAR, China; peterng.cuhk@gmail.com

**Abstract:** Why study China's Christian universities in the Republican era today? Christian universities were brought by Western missionaries and evolved as an educational system in China at the beginning of the 20th century. They were eliminated during the restructuring of the Chinese higher education system in the early 1950s; however, Deng Xiaoping's reform policies in the 1980s brought profound changes in China, encouraging Chinese scholars to bring back pre-1949 Christian higher education in China. Since then, new approaches and reconceptualizations have been developed, such as in the fields of Eastern–Western cultural exchange, interdisciplinary studies (from *xixue* to *guoxue*), and the adaptation of global and local perspectives. This paper is an attempt to report how the reconceptualizations of China's Christian universities in the Republican era were brought about in the various processes of indigenization, contextualization, internationalization, Asianization, and Sinicization, with the subsequent development of a new legacy moving toward the Sinicization of Christian universities.

**Keywords:** Christian universities; Republican China; indigenization; contextualization; internationalization; Asianization; Sinicization; global–local perspectives; glocalization

## 1. Introduction: Why Study Christian Universities in Republican China

Why study Chinese Christian universities in the Republican era today? Christian universities were brought by Western missionaries and evolved as a higher educational system in China at the beginning of the 20th century. They were eliminated during the restructuring of the Chinese higher education system in the early 1950s; however, Deng Xiaoping's reform policies in the 1980s brought profound changes in China, including the re-opening of the National Historical Archives (*Guojia Dangan Ju* 國家檔案局) which encouraged Chinese scholars to revisit and bring back Christian universities in China. Deng's "liberate thoughts" and "to seek truth from facts" became the basic guiding principles helping to encourage Chinese scholars to launch into the emerging field of the study of the history of pre-1949 Christian higher education in China with more open minds.

During the 19th century, Protestant missionaries came to China with the hope of modernizing China by introducing the Western method of education. They started building mission schools and grew them into Christian colleges in China. As a result, they were categorized as a significant part of the history of Christian missions in China (基督教在華宣教史). The re-study of pre-1949 Christian universities enlightened scholars to "seek truth from facts" and to see the history of Christian universities from a much broader perspective. They found that for the attempt to Christianize China to work, Christianity needed to adapt itself to Chinese culture and Chinese society and undergo various processes of glocalization in the Chinese context. Hence, in this paper, cases will be cited as examples of the reconceptualization of Christian universities in China seen in the various kinds of processes of glocalization developed in the study of Christian universities in China, such as indigenization, contextualization, internationalization, Asianization, and Sinicization.

## 2. Indigenization

The Boxer Movement was the beginning of the indigenization of Christianity in China.[1] According to the report by Jessie Lutz, there were very few Christian colleges in 1900, but 16 colleges were recorded by 1920. "There was a total of 199 college students in 1900 but the number reached as high as 1700 in 1920" ([Lutz 1971](), p. 102). Daniel Bays, discussing *The Growth of Independent Christianity in China (1900–1937)*, remarked that "the most important feature of this period was the growth of the spirit of indigenization in Chinese Protestant churches. This had barely begun in the nineteenth century, but it was a prominent theme after 1900" ([Bays 1996](), p. 308). Starting with the First International Symposium on the History of Pre-1949 Christian Higher Education held in Wuhan in 1989, Chinese scholars began to study the theme of indigenization in China's Christian colleges, such as the development of 'Chinese Studies Programs and Research in Christian Colleges' ([Tao and Ng 1998]()).

Indigenization is a process of bringing indigenous (local) knowledge into a foreign (such as Western) education system (such as in China's Christian colleges). Though it seemed to be paradoxical to have the study of *guoxue* (國學研究) in China's Christian colleges, Tao discovered that after the confrontation with the anti-Christian movement and the reclaiming of education rights in the 1920s, Christian colleges needed to speed up their processes of indigenization (ibid.). In those universities, a shifting of paradigms from the study of *xixue* (西學) to the study of *guoxue* (國學研究) was explicitly found, including the growth of Chinese Studies programs in universities such as Yenching University and Zhijiang (Hangchow Christian) University in China. Setting up a Chinese department (國文系) was a typical response to Western teachings, with more focus on retaining the traditional Confucian classics and teachings at Christian universities (K. [Zhang 2019](), pp. 289–316). Similarly, the establishment of the Harvard–Yenching Institute at Yenching University and the *guoxue yanjiu suo* (國學研究所) in other universities such as Cheeloo (Shantung Christian) University, the University of Nanking (Jinling), Central China (Huachung) University, West China (Huaxi) Union University, and Lingnan University were typical examples of this nationwide development in the 1920s and 1930s in China ([Tao and Ng 1998](), pp. 189–223). In addition to training Chinese scholars in the study of *Guoxue* ("國學", National Learning), the Harvard–Yenching Institute opened opportunities for Western scholars. Yenching University became a well-known training center for Sinologists such as Herrlee G. Creel, James R. Ware, Knight Biggerstaff, Derk Bodde, George E. Taylor, Arthur F. Wright, and Owen Lattimore, and it also became a platform for the study of Eastern–Western cultural exchange in China ([Tao and Ng 1998](), pp. 145–48; J. [Zhang 1992](), p. 41).

The rediscovery of Chinese studies programs in Christian universities reminded our scholars of Ma Xiangbo, a leading educator and the founder of Aurora (*Zhendan*) University (1903) and Fudan University (1905) in Shanghai and Fujen Catholic University in Beijing (1925). Ma introduced the study of Chinese classics and European—both Greek and Latin—classics, which included the study of languages and cultures, to these three Chinese universities, embracing the study of philosophy, religion, and Christian theology, as well as the role of religion in human life. Although, in the beginning, his work in Aurora and Fudan was not as successful, Ma eventually succeeded in offering a comprehensive curriculum covering both science and art subjects with a comparative, intercultural studies approach in early 20th century China ([Hayhoe 1988](), pp. 49–60; [Hayhoe and Lu 1996]()).

The existence of these Chinese studies programs in Christian universities was not paradoxical. Quite the contrary, they were seen as part of the processes of indigenizing Christian universities in China, providing richer resources for intercultural studies and setting up a platform for intercultural exchange in Christian universities. The best example is the establishment of the Harvard–Yenching Institute at Yenching University in 1928 during the high tide of the anti-Christian and anti-religious movements in China. Though it was short-lived, for just 10–20 years, it became a well-known research institute for Chinese language and cultural studies, specializing in training top-class Sinologists such as Herrlee G. Creel, Knight Biggerstaff, Derk Bodde, Arthur F. Wright, Owen Lattimore, and others ([Tao and Ng 1998](), pp. 145–48). Hence, the study of Chinese history and culture

in Christian universities turned out to be a legitimate part of the study of the history of Eastern–Western cultural exchange in 20th-century China.

### 3. Contextualization

Contextualization can be understood as a process of adaptation of Christian education to Chinese social and historical contexts. Zhang Kaiyuan, the pioneer of the movement of the study of China's Christian universities, set an example in the study of Miner S. Bates' papers (1897–1978). Bates was a Western missionary educator who taught history for thirty years at the University of Nanking. Bates' life and story were understood and retold by Zhang within a Chinese context, as a history teacher in China, and as an eyewitness of the Japanese "Rape of Nanjing". This application of the concept of contextualization resulted in the emergence of a brand-new approach, rediscovering Chinese history from a different, missionary's, perspective, yet offering a vivid picture of the study of modern Chinese history (Zhang 1996, pp. 1–27). Hence, Christian universities in China were brought back into the Chinese context and became an indispensable part of the history of the Nanjing Massacre in China. As a result, Zhang became a world-known scholar, both in the field of China's Christian universities and the study of the history of the Sino–Japanese War, through his discovery of Bates' writings about the Nanjing Massacre.

Another example is found in the study of Wu Yifang, the first Chinese president of Ginling College. The college motto of Ginling was "*Hou sheng* (厚生)," extracted from the saying of Jesus, which says, "I come that they might have life, and… more abundantly".[2] Wu expounded the Christian teaching within the Chinese social context by explaining that "Life… is rather to be able to help others and bring goodness to the society" (Zhu 2002, pp. 201–2). Wu attempted to contextualize the motto in a Chinese humanistic context by saying that "to live an abundant life was to serve the people", thus integrating Christian education into the formation of moral character in the Chinese socio-cultural context. It is interesting to note that Nanjing Normal University followed Ginling College by adopting the same concept of "*Hou sheng* (厚生). Nanjing Normal University has become famous for its nurturing of human persons (育人), a typical form of liberal arts education at the university. Nanjing Normal University has been appraised by the Ministry of Education as the 'national base of moral education' since 2003 and has become a worldwide center for the study of moral education.[3] Moreover, Wu Yifang Memorial Hall was set up in March 1987, and Ginling College was restored to commemorate Wu's work at Ginling (Waelchli 2008). Recently, another work in memory of Wu was written by Qian Huan Qi (錢煥琦), an alumnus of Nanjing Normal University and presently the Dean of Ginling College at NNU, to remember Wu's humanistic education in modern China (Qian 2014).

### 4. Internationalization

Internationalization is another way to reconceptualize the study of Christian universities in China. Ma Min wrote a paper in 1993 entitled "The Unique Characteristic of Internationalism in Christian Colleges- the case of Huachung University (華中大學), 1903–1952" (Ma 1996, pp. 74–110). His insight was broadened by seeing a Christian university as one that provided a platform for the study of Eastern–Western cultural studies, hence adding an international dimension to university education in China.[4] When Ma became the president of Central China Normal University in 2003, he started a new College of International Cultural Exchange (CICE) at the university. As stated in the college's information, CICE was founded in 1999 and inherited the legacy of a history of 100 years from *Huachung Daxue*. By the year 2010, CICE had embraced 1604 overseas students from 133 countries and regions (CICE 2014). This has added a new imperative for the development of modern higher education in China.

In 2002, Dana Robert, an American scholar, published an article on "The first Globalization (全球化): The Internationalization of the Protestant Missionary Movement Between the World Wars" (Robert 2002, pp. 50–67) affirming that the early Christian mission had been vested with a universal, globalized vision. In December 2003, a conference was held

by the Centre for the Study of Christianity of the Research Institute of World Religions, the Chinese Academy of Social Sciences in Beijing, with a theme entitled "Glocalization (全球地域化) and Christianity" (Zhuo 2004). Hence, in response to the Western concept of 'globalization', Chinese scholars called for a new concept of 'glocalization', which emphasized the expression of 'think globally, act locally' and the interplay between the processes of globalization and localization (Zhuo 2004, pp. 365–85).

Why the new concept of glocalization? According to Peter Beyer, Christianity has undergone a modern reconstruction since the Reformation era, which was a process of pluralization that turned Christianity into a drastically different form of 'multi-centred particularizations' (Beyer 2003, pp. 357–86). Beyer also argued that during the process of globalization, Christianity was confronted with other religions and turned out to be merely one of the many religions in the world, resulting in the loss of centralized authority within a variety of multi-centered, pluralistic versions of Christianity in different localities (ibid., pp. 367–69). Hence, the demand for a new concept of glocalization and the call for the study of any interplay between globalization and localization.

One significant example was found in the study of the formation of union universities in China in the early 20th century (Lazzarotto 2004, pp. 201–24). Christian schools spread throughout China, established by missionaries who came from various denominations or different mission societies. An idea for the amalgamation of mission schools was suggested in the *Chinese Recorder* in 1879.[5] Unfortunately, nothing could happen as the missionaries could not work together because they belonged to different denominations/mission societies with different educational ideals and mission policies. A typical example was found in Beijing, where the Methodist Peking University (匯文大學) tried to combine with North China College (潞河書院), but they could not because they shared varied mission policies and different theological traditions. It was only after the Boxer Movement that the missionaries could put aside their different traditions and merged to form new union universities in early 20th-century China. Eventually, a number of union universities emerged in the first two decades of the early 20th century in China, such as Shantung Union College (山東共和大學) in 1903, North China Union College (華北協和大學) in 1904, North China Union College for Women (華北協和女子大學) in 1906, North China Union Medical College for Women (華北協和女子醫學院) in 1908, West China Union University (華西協合大學) in 1910, Fukien Christian Union College (福建協和大學) in 1916, and Yenching University (燕京大學) in 1919 (ibid., pp. 218–23; Corbott 1955, pp. 69–70; Lutz 1971, pp. 106–15). It is interesting to note that the process of internationalization (globalization) of the mission schools in China turned out to be 'multi-centred particularizations', and they were eventually changed to become union universities by the Boxer outbreaks (the resulting processes of localization and indigenization) in China.

## 5. Asianization

In 2006, Chung Chi College of the Chinese University of Hong Kong celebrated the 55th anniversary of its founding in Hong Kong by organizing an international symposium on Christian higher education in Northeast Asia. For the first time, Chinese scholars attempted to extend the study of Christian higher education to a wider Asian regional context, and scholars from China, Japan, Korea, Taiwan, and Hong Kong were invited (Leung and Ng 2007). In addition to the introduction of the new concept of glocalization to the study of Christian higher education in different regions, scholars were encouraged to attempt comparative studies on issues of imperialism, nationalism, secularization, and modernization in the three countries—China, Japan, and Korea. It was fascinating that Christian higher education could be studied not only from intranational, but also from international contexts. As a result, a book was published entitled *Christian Response to Asian Challenges: A Glocalization View on Christian Higher Education in East Asia* (ibid.). Chinese scholars were also invited to join the North-East Asia Council of Studies of History of Christianity (NEACSHC), which is held bi-annually to promote the study of Christian higher education in Northeast Asian contexts. Since then, two books have been published.

The first was *Christian Mission and Education in Modern China, Japan, and Korea: Historical Studies* and the second was *Christian Presence and Progress in North-East Asia: Historical and Comparative Studies* (Jongeneel 2009, 2011).

A total of 48 papers were collected in the three volumes, and their significance lies in the attempts, especially for the Chinese scholars, to work together with Japanese and Korean scholars in applying the new concept of glocalization and in comparative studies on the development of Christian universities in the three countries. Some outstanding papers include Wang Licheng's "Globalization and Nationalism: A Comparative Study of Christian Higher Education in China and Korea" (Leung and Ng 2007, pp. 173–206); Mark Mullins' "The Challenges of Nationalism and Secularization for Christian Higher Education in Japan: Some Comparative Observations" (Leung and Ng 2007, pp. 207–26); Yuko Watanabe's "Christian Schools and Governmental Registration: Comparative Studies of Japanese and Chinese Christian Education" (Leung and Ng 2007, pp. 227–51); Peter Ng's "From 'Cultural Imperialism' to 'Cultural Exchange': Christian Higher Education in China Revisited" (Jongeneel 2009, pp. 43–53); Rui Kohiyama's "Women's Education at Mission Schools and the Emergence of the Modern Family in Meiji Japan" (Jongeneel 2009, pp. 99–114); Chun Chae-ok's "Rediscovering Ewha Mission and its Contribution to Education" (Jongeneel 2009, pp. 115–29); Zhuo Xinping's "Christianity and Contemporary Social Developments in North-east Asia: Reflections on the Future Development of Christianity in China" (Jongeneel 2011, pp. 20–25); Peter Ng's "Nationalism, Modernization, and Christian Education in 20th Century East Asia: A Comparison of the Situations in China, Japan, and Korea" (Jongeneel 2011, pp. 57–72); Park Jong-hyun's "The Holiness Mission and Church in Japan, Korea, and China (1901–1943)" (Jongeneel 2011, pp. 190–205).

In short, the Northeastern Asian contexts enlightened scholars to find that the development of Christian higher education was very different due to the different social, cultural, and political contexts in these three countries. Hence, an interplay between the global and local processes was found, and the impacts of "Asian challenges and the Christian responses" were seen more vividly (Leung and Ng 2007, pp. 1–20). For example, in Japan, although the Japanese government favored Western science culture, it was against the promotion of Christianity as a religion in Japan, whereas in Korea, especially during the Japanese colonial period, the colonial government wanted to control every aspect of Korean life. The government introduced Japanese Shinto shrine worship in all Korean schools. It was in this socio-political context that many Korean people turned to Christianity as a means to promote Korean nationalism against Japanese imperialism, a very distinctive role of Christianity in Korea. In contrast, China was very different from both Japan and Korea. Compared with Japan and Korea, the Chinese government adopted a more lenient attitude toward Western Christianity. China's Christian universities turned out to provide a platform for dialogue between Chinese nationalism and Christian internationalism, and Christian higher education in China helped to enlighten the process of modernization in China (Jongeneel 2011, pp. 57–72). For example, Yenching University made serious efforts to provide a platform for a dialogue between Christianity and Chinese culture, which could enhance intercultural and interdisciplinary studies. Hence, another vivid example of the interplay between globalization and localization in China's Christian universities (Ng and Ng 2014, pp. 49–58).

## 6. Sinicization

In the 1980s, Paul Cohen, a student of John Fairbank, attempted to go beyond Fairbank's "impact–response" paradigm and proposed a "China-centered approach" to the study of modern Chinese history (Cohen 1984). Daniel Bays adopted this China-centered approach to the study of Chinese Christianity and China's Christian universities, collecting valuable papers for his two books, *Christianity in China: From the Eighteenth Century to the Present* and *China's Christian Colleges: Cross-cultural Connections 1900–1950* (Bays 1996; Bays and Widmer 2009). Although Bays succeeded in putting China at the center and focused more on the study of Chinese Christians (Bays 1996, p. ix), China's Christian colleges

were still studied from a distinctively American context and perspective (Bays and Widmer 2009, pp. xiii–xx). At the 2019 conference held in Wuhan, Elizabeth Perry, the Director of the Harvard–Yenching Institute (HYI), retold the history of HYI from a more global–local perspective and reported that, in addition to the introduction of liberal arts education in China, HYI had put much emphasis on the study of Chinese classics, literature, and Chinese history, training Chinese scholars such as William Hung (Hong Ye), Qian Mu, Feng Youlan, Qi Sihe, Chen Yuan, Rong Geng, and others (Perry 2022, pp. 5–8).

In more recent years, the concept of Sinicization has evolved as a popular topic among Chinese scholars in the study of Chinese Christianity (Zhang 2016, p. 21; 2017, pp. 50–53; Zhuo 2017, p. 42; Tang 2017).[6] Chinese scholars also began to retell the story of China's Christian colleges by locating them in the Chinese historical context and perspective, such as Peter Ng's "Resurgence of Study of China's Christian Higher Education since the 1980s" (Ng 2019, pp. 364–86), and Ma Min's "The Fusion of Chinese and Western Cultures—Revelation from the Study of China's Christian Universities" (Ma 2022, pp. 13–29). In retelling the story of China's Christian universities, Ng gave an account of the implementation of Deng's reform and opening policy as a distinctive impulse for the resurgence of the study, exemplifying it as one way that Sinicization reconceptualizes the study of China's Christian universities from the Chinese historical context. In contrast, Ma adopted the concept of Sinicization and strengthened it with a forward-looking approach, proposing that the study of Christian universities could provide scholars with new insights for the modernization of Chinese higher education, including (a) the emphasis on quality education, (b) paying more attention to internationalization and the fusion of Chinese and Western cultures, (c) the provision of quality teaching and training of high-quality scholars, and (d) the construction of effective systems and harmonious environments on university campuses (Ma 2022, pp. 13–29). The study of Christian universities in China now conforms to the proper direction toward which Chinese society is heading (the Chinese social context) and is becoming a legitimate part of the modern higher education system in China. Hence, the emergence of reconceptualization of the legacy of China's Christian universities, with a higher aim to seek enlightenment for the advancement of modern higher education in China.

## 7. Concluding Remarks

There has been a resurgence of the study of China's Christian higher education since the implementation of Deng Xiao-ping's reform and opening policy, following Deng's motto of "liberate thoughts" and "seek truth from facts". In 1989, Zhang Kaiyuan organized the First International Symposium on the History of Pre-1949 Christian Universities in China at Central China Normal University (CCNU) and called for new approaches to the study of China's Christian universities, such as seeing them as "a platform for Sino–Western cultural exchanges" (Zhang and Waldon 1991, p. 2). Zhang also encouraged scholars to equip themselves with "conscious commitment and social responsibility, not only to write, but also to create (a new understanding of) history, for the good of the society and the human world" (Ma 2006, p. 420). In 2019, Zhang and Ma worked together to organize another conference at the same university (CCNU), entitled "Retrospect and Prospect: The International Symposium on Thirty Years' Research on History of Christian Colleges in China", calling together four generations of doctoral students they have trained and celebrating their work over the past 30 years. At the conference, Ma affirmed that the legacy of Christian colleges in China had been reconceptualized and has eventually become a legitimate part of the Chinese higher education system (Ma 2022, pp. 13–29). With the new understanding of Sinicization, the study of China's Christian universities changed to become truly "China's Christian universities", and a legitimate part of the modern higher education system in China.

Take Yenching University again as an example. John Leighton Stuart (1876–1962) recalled his ideals of founding Yenching University and noted that "Yenching University should become a Chinese university and people would only be reminded of its Western

origin when recalling its founding history" (Stuart 1946). It was during the Sino–Japanese War that Christian universities and their presidents were tested. Would they 'Aiguo aijiao' (愛國愛教, 'love country and love religion') or be Sinicized? Surprisingly, Christian universities demonstrated their *Aiguo* through their outstanding services given to the nation during the war (Hu 2000; Liu and Liu 2003). Hu Shi wrote several letters and speeches to the United Board for Christian Higher Education office in New York, affirming and praising China's Christian universities for their 'remarkable spirit of heroism and self-sacrifice' and their serving in the wartime as 'centers of national life and places for training of patriotism… "(hence,) we (the Chinese government) welcome and in fact have great need of these schools'' (ABCCC 1941, p. 8). In this way, Hu was speaking on behalf of the Chinese (Republican) government and affirming that Christian universities in China had already become loving-China (*Aiguo*) Christian universities (i.e., "truly China's Christian universities"), which was especially apparent during the Sino–Japanese war.

In early 1951, Mao Zedong, the Chairman of the New China government, offered to write the Chinese characters of *Yanjing daxue* ("燕京大學") on the western gate of the university as a significant sign of his high regard for the work of Yenching University in Beijing since its inauguration and during the civil war (Chen 2013, p. 218).[7] It could also be understood as a significant sign to affirm that Yenching University was already recognized as a Chinese university by the new (Communist) government.[8]

Although the New Chinese government had tried to follow the Soviet model of social determinism and the promotion of professional education in the 1950s, China had to turn back to a more comprehensive, humanistic approach to education, including liberal arts education, under the "reform and opening" policy during the 1980s (Sun 2018, pp. 60–66). In 2007, a new college was set up at Peking University, Yuan Pei College (元培學院), honoring Cai Yuanpei's educational ideals of "liberality, democracy and all-inclusiveness" (思想自由、兼容並包), with more emphasis on rebuilding the liberal arts tradition in the university. A new name, "humanistic quality education" (人文素質教育), was employed, which was close to the liberal arts education models introduced by the Christian universities a century ago.[9] The revived liberal arts education ("*Boya Jiaoyu*", 博雅教育) was kept, where "*Bo*" (博) means being erudite, and "*Ya*" (雅) means nurturing decency. Hence, it is more than extensive learning; it also involves embracing integrity, compassion, and spirituality and nurturing students in whole-person education (Sun 2015, pp. 15–21; 2018, pp. 60–66; Mou 2021, pp. 73–88; Tong 2023, p. xiii).

In 2008, a new research institute, the Institute of Advanced Humanistic Studies (IAHS), was established, and an effort was made to rekindle collaboration between Peking University and Harvard University and the world, with blessings from the heritage and good spirit of the Harvard–Yenching Institute (YUAN 2014, p. 48). As the history of Yenching was fully recognized, the Yenching Center and Yenching Academy were both established in 2014, again as symbols of reviving the global mind and spirit of Yenching University at Peking University (YAPU 2016, 2022; YCPU 2022; Rosenbaum 2015). Also in 2014, an international conference was jointly organized by Peking University's IAHS and the Beijing Alumni Association of Yenching University (BAAYU), entitled "Yenching University and Liberal Arts Education Tradition in Modern China".[10] It was affirmed that the tradition of liberal arts education in modern China was closely related to Yenching University, and more significantly, the BAAYU was officially invited to this joint venture while they were celebrating the 95th anniversary of the founding of *Yenda* (1919–2014). Hence, a legitimate status was given to *Yenda* and the BAAYU, and the history of *Yenda* as a legitimate part of China's university education system was re-established (YUAN 2014, p. 48). It was also exciting to note that the BAAYU celebrated the 100th anniversary of the founding of *Yenda* inside the *Beida* campus on 8 October 2019, with Prof. Isabel Crook (饒素梅) as one of the honorable guests. Crook was a recipient of China's top honor for foreigners, the Medal of Friendship, awarded by President Xi Jinping (Qu 2019).

The resurgence of the study of China's Christian universities since the 1980s has brought new representations and reconceptualizations of the legacy of Christian univer-

sities in China. The global–local approach and the glocalization perspective enabled scholars to rediscover that the study of China's Christian universities and the development of reconceptualization was moving toward the Sinicization of Christian higher education in China. China's Christian universities have now become more properly understood in the processes of indigenization, contextualization, internationalization, Asianization, and Sinicization, so much so that China's Christian universities have now become truly "loving-China's (*Aiguo*) Christian universities". This precisely echoes what Dr. Gu Ziren (Koo T. Z. 顧子仁, 1887–1971) had said about the fate of foreign organizations such as the YMCA in China, namely "it would eventually be 'of the people, by the people, and for the people'" (Koo 1924, p. 45; Yang 2022).

**Funding:** This research received no external funding.

**Data Availability Statement:** Data and references are contained in the article.

**Conflicts of Interest:** The author declares no conflict of interest.

## Notes

[1] The statement was first made by a Protestant historian, Wang Zhixin (王治心) who commented on the significance of the Boxer Movement, saying: "Protestant Christianity changed very much after 1900." (Wang 1979, p. 242). David Buck, an American scholar shared similar view in saying that: "Boxer Movement is not only a turning point in this century, it is also a divide splitting two different responses from traditional China and modern China toward foreign imperialism." (Buck 1987, p. 6). A Japanese historian, Sumiko Yamamoto also affirmed it with the rapid growth of Christians in China from 1900–1917, saying that "number of Christian converts rose from 112,808 in 1900 to 312,970 in 1917". (Yamamoto 2000, pp. 21–22) Leung Yuensang (梁元生) moved even further to claim that "The unique perspective of the 'victums' (referring to the Chinese Christians) has potentially developed into a stream of 'Christian Historiography' (基督教史學) which would probably arouse an equal attention as 'Boxer Historiography' (義學)." (Leung 2001, pp. 536–44). In June 2004, at the International Conference on the Boxer Movement and Christianity in China, there was a research paper reported the findings of some distinctive scenarios of the impact of Boxer Movement on the work of Christian Higher Education in China, including the founding of Shanxi University by Timothy Richard, the establishment of Yale Mission in China (later known as Yale-China Association), and the emergence of union Christian universities in the 1900–1920s. (Lazzarotto 2004, pp. 202–24).

[2] A quotation from *The Gospel of John* 10: 10.

[3] Nanjing Normal University had successfully organized an international conference in 2011, on the theme entitled: "Cultivating Morality: Human Beings, Nature and the World,", hosting 300 participants from 33 countries. The conference indeed affirmed her leadership status on the field of moral education in China. See e.g., (Taylor 2011).

[4] He has rightly echoed with the call of Zhang Kaiyuan to broaden the study of China's Christian universities from the perspective of China-Western cultural studies. See, e.g., (Zhang and Waldon 1991; Zhang 1996).

[5] The issue of "amalgamation of mission schools" first appeared on *Chinese Recorder* in November 1879. (Baldwin 1879, pp. 456–57; also Lutz 1971, pp. 104–5). The issue was raised again by the Chairman of the Conference Education Committee, Dr. F.L. Hawk Pott (卜舫濟), with the proposal of establishing an "Inter-denominational Union Christian University" in China. (China Centenary Missionary Conference Committee 1907, pp. 70–75). One missionary even recalled, saying: "Up till the Boxer years, though the friendliest relations always obtained between the missionaries of the Baptist Mission Society and those of neighboring missions, each had its own distinct field and carried on its own evangelistic and educational work. But God… over-ruled the Boxer outbreak to bring the workers of the various societies closer together and gave them the chance to plan new enterprises in cooperation…They were thus …led to review the whole situation and plan united for the future." (Corbott 1955, p. 63).

[6] Western scholars are arguing that there are two kinds of Sinicization in China, one is "Sinicization from above," i.e., from the Party-State and government; and the other is "Sinicization from below," namely from scholars, religious leaders or lay believers. (Madsen 2021, pp. 1–15) The one from above was then classified as "Chinafication" which could somewhat be distorted by its political intentions to regulate religions. (Yang 2021, pp. 16–43). But Chinese scholars turned out to be presenting a more positive and forward-looking approach, as reported.

[7] As reported by He Di (何廸) in his paper, "Yenching University and the Modernization of Chinese Education", presented at the First International Symposium on the History of Pre-1949 Christian Universities in China, held at Central China Normal University, Wuhan in 1989. See also (Ng 2014, p. 77).

[8] It was at the same time (Spring of 1951) that Mao issued his order to appoint Lu Zhiwei (陸志韋) to be the President of Yenching University. See, (Chen 2013, p. 225).

⁹   The development of "Humanistic Quality Education" was a significant move in China. Some scholars have interpreted it as "Cultural Quality Education", but the term "Humanistic" (in Chinese =人文) goes far beyond the meaning of "Culture" (文化). See (Ng et al. 2023, pp. 13–14, 201–2).

¹⁰   The 2014 conference was organized with a theme entitled: "Yenching University and Liberal Arts Education Tradition in Modern China". It not only affirmed that the tradition of liberal arts education in modern China was closely related to Yenching University, but also connected it with and confirmed what President Cai Yuanpei proposed when he took up the leadership of the university with the motto of "learning from both China and the West, thinking freely, and being inclusive.".

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
