# Peer review of "Reconceptualizing the Study of Christian Universities in the Republican Era in Today’s China"

_religions, doi:10.3390/rel15010103_

Round 1
Reviewer 1 Report (Previous Reviewer 4)
Comments and Suggestions for Authors
The author has made significant revisions to the paper addressing many of the issues from my previous review. Here are the main improvements:
1. The abstract clearly summarizes the key points and purpose of the paper now.
2. The introduction provides more context and background on why the study of Christian universities in Republican China is important. This helps motivate the rest of the paper better.
3. The author has expanded on the different processes of re-conceptualization such as indigenization, contextualization, etc. and given more examples to illustrate each concept. This adds valuable detail and support to the main arguments.
4. The conclusion now neatly ties together the various threads of argument and shows how the re-conceptualization of Christian universities in China has led to them becoming truly "Chinese" institutions recognized as part of China's educational system. The Yenching University example reinforces this effectively.
Overall, the revised paper reads much more coherently and makes a persuasive case about how views on Christian universities in Republican China have evolved. The added examples and explanation of key concepts strengthens both the scholarly contribution as well as the narrative flow. I think with these significant improvements, this paper makes a fine contribution worthy of publication after some minor editing and proofreading.
Author Response
Thanks for the reviewer’s affirmation of all the improvements made. Thanks also for the additional remarks which are helpful to tie up the whole paper.
Reviewer 2 Report (Previous Reviewer 3)
Comments and Suggestions for Authors
It appears that the author lacks comprehensive knowledge in religious studies and Chinese studies within the contemporary academic landscape. Moreover, it would be prudent for the author to refrain from using the potentially problematic term "Sinicization." Furthermore, the use of the term "contextualization" without proper reference to a major scholarly reference is also highly imprudent.
The field of "Christian universities in Republican China" has been widely explored over the past seventy years, with numerous articles and books contributing to its discourse. Therefore, the most effective way to present this article would be to augment its current title with "in Today’s China." A more fitting title for this article would be "Reconceptualizing the Study of Christian Universities in the Republican Era in Today’s China." This revised title is expected to enhance the article's overall presentation.

Author Response
Thanks to the attentive reading of this paper. The five keys concepts, “indigenization “, “contextualization,” “internationalization,” “Asianization, “and “Sinicization“ represent the different processes of re-conceptualization and the various spheres of studies in which Chinese scholars have been working on China’s Christian universities under the theme of glocalization in the past three decades. And the conclusion ties together the various threads of argument and shows how the re-conceptualization of Christian universities in China has led them to becoming truly "Chinese" institutions recognized as part of China's educational system. The example of Yenching University reinforces this effectively.
Reviewer 3 Report (New Reviewer)
Comments and Suggestions for Authors
The manuscript discusses the perspectives of the study of Christian universities with new approaches and re-conceptualizations. The authors analyzed the research of Christian universities in the past thirty-four years (from 1989 to 2023).
Please clarify the following suggestions for publication.
The authors discuss “in training top-class Sinologists,” please provide unique examples that will give the readers a better understanding (p.2). Do the study of the training of Sinologists concerning the study of Chinese Christians (p.6)? Are there any differences between the study of Guoxue and Hanxue in the Christian universities in the Republican Era? (p.2)
The authors state, “Boxer Movement was the threshold of the indigenization of Christianity in China” (p.2), and the Christian universities “were eventually changed to become union universities by the Boxer outbreaks in China” (p.4). Please discuss more clearly the impact of the Boxer Movement on the development of Christian universities in Republican China.
Please provide the reference source for the discussion on “an idea of amalgamation of mission schools suggested in the Chinese Recorder in 1879” (p.4).
The paragraph “There was a total of 48 papers collected in the three volumes ... ... Church in Japan, Korea, and China (1901-1943)” (p. 5) does not show any significant discussion.
Are there any relations among the “indigenization “n,” “contextualization,” “internationalization,” “Asianization, “and “Sinicization “in the study of Christian universities in China?
Author Response
Thanks so much for this reviewer’s report. All the questions raised reflect careful readings of the text and thoughtful queries. My responses are as follows:
- The two terms Guoxue and Hanxue should be distinguished and separated. Guoxue (國學) is “National Studies”, and the study of Guoxue in China’s Christian Universities seemed to be a strange phenomenon, yet eventually it became an unique and outstanding phenomenon in China’s Christian universities. The study of Hanxue (漢學) is Sinology, which was another distinctive phenomenon at China’s Christian universities too. Yenching University was an outstanding example found in China’s Christian universities. Hence, clarifications were made in lines in 59-80.
- Regarding the statement, “Boxer Movement was the threshold of the indigenization of Christianity in China”(p.2) and the Christian universities “were eventually changed to become union universities by the Boxer outbreaks in China” (p.4), The first statement was made by a Protestant historian, Wang Zhixin (王治心) who commented on the significance of the Boxer Movement, saying: “Protestant Christianity changed very much after 1900.” (Wang, 1979, 242). David Buck, an American scholar shared similar view in saying that: “Boxer Movement is not only a turning point in this century, is also a divide splitting two different responses from traditional China and modern China toward foreign imperialism.” (Buck, 1987, 6). Leung Yuensang (梁元生) moved even further to claim that “The unique perspective of the ‘victums’ (experienced by the Chinese Christians) has potentially developed into a stream of ‘Christian Historiography’ (基督教史學) which would probably arouse an equal attention as ‘Boxer Historiography’ (義學).” (Leung, 2002, 536-544). Following this line of thought, Ng Tze Ming (吳梓明) launched his research and discovered some distinctive scenarios of the impact of Boxer Movement on the work of Christian Higher Education in China, including the emergence of union universities in the 1900-1920s. (Lazzarotto et al., 2004, 202-224). Clarification is made by at the endnote, please refer to the first endnote added on page 8.
- Referring to the question on “an idea of amalgamation of mission schools suggested in the Chinese Recorder in 1879” (p.4), the issue was raised by the Chairman of the Conference Education Committee, Dr. F.L. Hawk Pott (卜舫濟), with the proposal of establishing an “Inter-denominational Union Christian University” in China. ( China Centenary Missionary Conference Committee, 1907, 70-75) . One missionary even recalled, saying: “Up till the Boxer years, though the friendliest relations always obtained between the missionaries of the Baptist Mission Society and those of neighboring missions, each had its own distinct field and carried on its own evangelistic and educational work. But God… over-ruled the Boxer outbreak to bring the workers of the various societies closer together and gave them the chance to plan new enterprises in cooperation…They were thus …led to review the whole situation and plan united for the future.” (Corbott, 1955, 63). A fifth endnote is added on page 8 for clarification.
- Regarding to the question about the significance of “There was a total of 48 papers collected in the three volumes ... Church in Japan, Korea, and China” (p. 5), their significance lies on the attempts of scholars, especial Chinese scholars, to apply the concept of glocalization and to work for comparative studies on Christian universities found in China, Japan and Korea. Their significance is spelt out now. Please refer to lines 214-217.
- Finally, regarding the question about the relations among the five key concepts in the study of Christian universities in China, the reply is: they all represent the different processes of re-conceptualization and the various spheres of studies in which Chinese scholars are working on China’s Christian universities under the theme of glocalization in the past three decades. And the conclusion ties together the various threads of argument and shows how the re-conceptualization of Christian universities in China has led them to becoming truly "Chinese" institutions recognized as part of China's educational system. The example of Yenching University reinforces this effectively.
Thanks again for all the careful readings of this paper.
This manuscript is a resubmission of an earlier submission. The following is a list of the peer review reports and author responses from that submission.
Round 1
Reviewer 1 Report
Comments and Suggestions for Authors
This is a thought-provoking paper that gives an excellent analysis of various dimensions of the contribution of China’s historic Christian universities to the deepening of inter-cultural understanding between China and the Western world in China’s republican period. The fact that it is based largely on research published by scholars in China over the past two decades brings new perspectives and considerable authenticity to the argument around re-conceptualization of their contribution. I definitely support publication of this piece and just have one main suggestion, as well as one minor correction. The concept of glocalization is mentioned in the introduction, in connection with indigenization, contextualization, internationalization, Asianization and Sinicization. The paper would benefit from some comment on how the term glocalization has emerged from studies of globalization in recent times, and then some discussion of the related set of concepts that are used to organize the paper. In turn it would be good to see some kind of bridging, from one concept to the next in the subsequent main sections of the paper. Beginning with indigenization and then contextualization makes good sense, but why is internationalization placed before Asianization, and then why is Sinicization placed last? The paper could be improved by a more compelling flow from one concept to the next, some careful thought about the ordering of the concepts and some explanation that will enable the reader to be carried from one to the next.
The work of Ma Xiangbo is briefly referenced in the introduction and his efforts to integrate Chinese and Western classical knowledge in the curriculum of the early universities he founded. In fact this was really only successful in the case of Fujen University, not the earlier Zhendan and Fudan universities – a minor revision of the text would be appropriate here.
Apart from these minor points, this is an excellent paper that makes a significant new contribution to the understanding of higher education history in China’s Republican period and also the ways in which this reconceptualization is contributing to current reform initiatives in Chinese higher education.
Comments on the Quality of English LanguageThe quality of English expression is okay - only minor adjustments by a first language English editor would be helpful.
Author Response
Thanks for the reviewer’s overall comments and the affirmation that this paper would make a significant contribution to boarden the understanding of the history of China’s Christian universities in the Republican period. And all constructive proposals on this paper are noted.
The paper is now revised accordingly, especially to the suggestions of giving more background information regarding how the concept of glocalization was emerged as a Chinese response to the Western conception of globalization in the study of Christian mission in China, clarifying the flow of changes and bridging the connections between the different concepts of glocalization used. (See revised paper esp. lines 145-166) The remarks on Ma Xiangbo are noted and clarified too. (See revised paper lines 82-85).
Reviewer 2 Report
Comments and Suggestions for Authors
Christian universities, a modern higher education system, was brought to and flourish in republican China because of the efforts of Western missions and many Chinese organizations and celebrities. As an component of the modernizing process, this educational system without any doubt has had great influence on every aspect of China's politics, society and diplomacy. A hitory without Christian universities wouldn't be sufficient for the understanding of modern China. In this sense, Christian universities in Republican China deserve a thorough research.
This manuscript attempts to make a comprehensive review of several approaches in scholarship, namely, indigenization, contextualization, internationalization, Asianization and Sinicization. In addition, the author also points out the legacy of Christian universities on today's Peking University and such.
Regretably, scholars who are familiar with Chinese Christian history wouldn't found any new knowledge through this manuscript. In other words, this manuscript fails to demonstrate its originality. There have been extensive scholarly works on China's Christian universities published in both Chinese and English. Many of Chinese works are referenced here but many English publications that are revelant and important are absent. This might be a reason that prevents the author from knowing the recent developments in this field.
Comments on the Quality of English LanguageLanguage is acceptable at this stage. However, it does need moderate editing, especially those paragraphs that read like an obvious translation from Chinese text.
Author Response
Thanks for the reviewer’s affirmation that the study of China’s Christian universities in the Republican Era deserves a modern and thorough research. Yes. And all the suggestions are constructive to this paper and were well-noted too.
This paper has now been revised drastically to make it a thorough one as suggested. The originality of this paper is precisely the attempt to analyze the legacy and the study of China’s Christian universities from a glocalization perspective, with the five key concepts introduced which have not been spelt out as clearly before, and citing as extensively works from global scholars too. (See revised paper lines 32-44 & 145-164) For example, there have been 3 conferences held by North-east Asian scholars, in Hong Kong (2006), Seoul (2007) and Wuhan (2009), with special attention to the development of Christian higher education in North-east Asian contexts. Hence, more references are cited from the work done by the Japanese, Korean and Western scholars in this revised paper, to report more extensively the recent development of works relevant to the study of China’s Christian universities, and to justify the contemporary thinking of the interplay between the global and local processes and perspectives discussed in this paper. (See revised paper lines 187-245)
Reviewer 3 Report
Comments and Suggestions for Authors
The current draft (Reconceptualizing the Study of Christian Universities in the Republican Era) is superficial and incomplete. The reader was expecting a well-researched and in-depth analysis on the historiography of the study of Christian universities in the first half of the 20th century. However, the paper failed to contribute anything meaningful to the current field. A real contribution should come from a careful review/comparative studies on the academic works of this topic in China, Hong Kong, Taiwan, as well as other parts of the world. Based on this consideration, the author needs to thoroughly revise the current draft.
Another serious error is that the author failed to do an adequate literature review. Nowadays, there have been hundreds, if not thousands, of articles and books dealing with Christian universities in China. The author apparently fails to catch the trend and major features of the Christian universities in the late Qing and Republican periods in China. For example, a conference proceedings entitled《回顾与展望——中国教会大学史研究三十年》(co-edited by 章开沅、马敏 and Elizabeth J. Perry) was published last year. Another example comes from the late Professor Daniel H. Bays and Professor Ellen Widmer, who co-edited a book entitled China's Christian Colleges: Cross-Cultural Connections, 1900-1950, published in 2009. It is hard to believe the author left out the pioneering work of this field, China and the Christian Colleges, 1850-1950, by the late Professor Jessie Gregory Lutz, who just passed away last December. How could the author ignore the major and modern publications while dealing with a topic in this field?
In addition, the author needs to reexamine his scholarly commitment. The re-opening of National Historical Archives (Guojia Dangan Ju 國家檔案局), as mentioned by the author, did help Chinese scholars to conduct research on Christian universities. However, if the author really understands the later-on condition in China, the conservative attitude of the national and provincial historical archives has already curbed the possible development of Christian studies. Anyone who wants to proceed with research on Christian universities, he/she sure understands how difficult it is to conduct research or to publish academic work in this field. The author should adopt Deng Xiaoping’s spirit of “To seek truth from facts,” to point out the current adverse situation in the study of Christian universities in China. This article, in its current form, seems to lean to one side without being faithful to the real difficulty of the academic situation in China.
Comments on the Quality of English Language
One of the gravest errors of the paper is the author simply plays with some terms without any solid contributions. Those terms like indigenization and contextualization have been used for a long time and a great part of this article has contributed nothing new to the scholars of this field. Furthermore, the author fails to offer clear definitions of key terms in this article. Some terms, like Sinicization and Asianization, are ambiguous and problematic while others, like indigenization and contextualization, have been questioned by some theologians in their application to church studies. Therefore, it is crucial to define those key terms at the beginning of this paper.
There are quite a few language problems in the article. Even the first sentence of the Abstract is problematic (Abstract: Why the study of China’s Christian Universities in the Republican era?). The author might need a native speaker to carefully proofread the paper and remove/polish all sentences in question.
Author Response
Thanks for all the suggestions which are very constructive regarding the improvement of the first paper. Revisions were made extensively to include all possible references the reviewer requested. Yes, the study of the history of China’s Christian universities in the Republican Era would not be an easy field of study without the groundwork laid by Prof. Zhang Kai Yuan and others since 1989. And the study had gone through steadily in the past 30 years. The 2019 conference held at Central China Normal University, Wuhan and the subsequent conference proceedings co-edited by Zhang et al in 2022 could give scholars a comprehensive and an excellent report of the 30 years’ research on History of Christian Colleges in China (1989-2019). Now, three distinctive papers are added to clarify the more recent developments and to justify the use of the term “Sinicization” in the final stage. The three papers were given by Perry Elizabeth, Peter Ng and Ma Min at the 2019 conference cited for reference. (See this revised paper, lines 255-284) Also, the work of Jessie Lutz who had attended the 1989 conference, and of Daniel Bays et al were recalled in this paper too, as they were the pioneers who started their research on Chona’s Chirstian universities in the 1970s and 1990s. (See this revised paper, lines 47-53 & 247-255) Though the field of study did not go through as smoothly around the turn of the century, and there were difficulties found. However, an international conference could still be held in 2019 at Wuhan, with the gathering of those pioneers who attended the 1989 conference, together with the 3-4 generations of their doctoral students testified that the faithful ones could overcome any difficult situations in China. (See this revised paper, lines 295-304) Nevertheless, thanks again for all the helpful suggestions.
Reviewer 4 Report
Comments and Suggestions for Authors
This article provides a thoughtful overview of the reconceptualization of Christian universities in Republican era China. The author highlights how perceptions and approaches to studying these universities have evolved since the 1980s as a result of Deng Xiaoping's reforms. Specifically, the article examines how concepts like indigenization, contextualization, internationalization, Asianization, and Sinicization have been applied to re-frame the legacy of Christian higher education in China.
The article emphasizes that Christian universities are now studied less as part of Christian missionary history in China, and more as a chapter in the development of modern Chinese higher education and East-West cultural exchange. The varied processes of "glocalization" experienced by Christian universities are insightfully analyzed through examples like the rise of Chinese studies programs, changing curricula, and regional vs global orientations. The article concludes by relating the reconceptualization of Christian universities to the revival of liberal arts education in contemporary China.
This is a clearly organized, well-researched contribution that sheds light on an important dimension of the history of higher education in modern China. By tracing changing historical appraisals of Christian universities, it offers a nuanced perspective on both China's reception of Western knowledge and the integration of foreign institutions.
This article will be of interest to scholars of Chinese education, history, religion and Sino-Western relations.
Here are a few suggestions that could potentially improve this article:
1. The article would benefit from more explicit definitions of key concepts like indigenization, contextualization, etc. Clarifying what the author means by each term would make the analysis more precise.
2. More concrete examples illustrating how Christian universities embodied these concepts would strengthen the discussion. The paper currently relies heavily on asserting these processes occurred rather than demonstrating them.
3. The transnational and intercultural dynamics noted in the conclusion could be integrated and developed further in the main discussion rather than just at the end. Expanding on these ideas earlier would reinforce the cross-cultural approach.
4. The article focuses primarily on Protestant universities. Including some discussion of Catholic institutions would offer a more comprehensive picture of Christian higher education.
5. More attention could be paid to limitations and challenges faced by Christian universities in enacting glocalization, not just successes.
Author Response
Thanks so much for the reviewer’s affirmation of value of this paper towards the reconceptualization of and the understanding of the history of China’s historic Christian universities in the Republican period. And all the suggestions are helpful and constructive to the revision of this paper.
Accordingly, (1) the five key concepts, namely indigenization, contextualization, internationalization, Asianization, and Sinicization are further clarified and put more closer into the social-historical contexts. (2) More examples and references are cited. (3) The international and cross-cultural elements were added too, (See changes throughout the paper). (4) The reference to Ma Xiangbo’s three colleges, namely: Aurora University, Fudan University and Fu Jen Catholic University were examples of Christina (Catholic) higher education in China. Ma Xiangbo was a Catholic priest, and a pioneer educator in the introduction of the study of Greek and Latin Classics, hence a cross-cultural and trans-national classical study into Chinese higher education. This is the most distinctive feature found in Catholic higher education in China too. (See revised paper lines 76-86). (5) Several papers were cited from the conference proceedings from Hong Kong (Leung & Ng, 2007), and Japan and Wuhan (Jongeneel et al, 2009 &2011) reviewed and discussed challenges of Glocalization upon Christian universities, and one example has been quoted for reference too. (See revised paper lines 188-245) Thanks again for all the helpful and constructive suggestions.